# Identification and Toxicity Prediction of Biotransformation Molecules of Organophosphate Flame Retardants by Microbial Reactions in a Wastewater Treatment Plant

**DOI:** 10.3390/ijms22105376

**Published:** 2021-05-20

**Authors:** Yeowool Choi, Sang Don Kim

**Affiliations:** 1Convergence Technology Research Center, Korea Institute of Industrial Technology (KITECH), Ansan 15588, Korea; 2School of Earth Sciences and Environmental Engineering, Gwangju Institute of Science and Technology, 123 Cheomdangwagi-ro, Buk-gu, Gwangju 61005, Korea; 3Center for Chemicals Risk Assessment, Gwangju Institute of Science and Technology, 123 Cheomdangwagi-ro, Buk-gu, Gwangju 61005, Korea

**Keywords:** organophosphate flame retardant, biotransformation, wastewater treatment plant, microorganisms

## Abstract

Organophosphate flame retardants (OPFRs) are substances added to plastics, textiles, and furniture, and are used as alternatives to brominated flame retardants. As the use of OPFRs increases in the manufacturing industry, the concentration in the aquatic environment is also increasing. In this study, OPFRs introduced into a wastewater treatment plant (WWTP) were identified, and the toxicity of biotransformation molecules generated by the biological reaction was predicted. *Tris*(2-butoxyethyl) phosphate, *tris*(2-butoxyethyl) phosphate, and triphenyl phosphate were selected as research analytes. Chemicals were analyzed using high-resolution mass spectrometry, and toxicity was predicted according to the structure. As a result, *tris*(1-chloro-2-propyl) phosphate showed the highest concentration, and the removal rate of OPFRs in the WWTP was 0–57%. A total of 15 biotransformation products were produced by microorganisms in the WWTP. Most of the biotransformation products were predicted to be less toxic than the parent compound, but some were highly toxic. These biotransformation products, as well as OPFRs, could flow into the water from the WWTP and affect the aquatic ecosystem.

## 1. Introduction

Organophosphate flame retardants (OPFRs) are substances applied when manufacturing plastics and fibers, and their production is gradually increasing due to the regulation of brominated flame retardants used in the past [1,2]. OPFR usage worldwide rose from 500,000 tons in 2011 to 680,000 tons in 2015, an increase of 36% over five years [3]. OPFRs are likely to be easily eluted in water as they are added without chemical bonding in the product [4]. Since OPFRs are used in various products, large amounts of OPFRs can be found in the aquatic environment. Recently, research on OPFRs has been conducted with a focus on surface water and wastewater treatment plants (WWTPs). The total concentration of OPFRs was 877 ng/L (28.3–16,000 ng/L) in Lake Shihwa, South Korea. An OPFR concentration of 164–627 ng/L was detected near the WWTP runoff. The main OPFRs detected were *tris*(2-butoxyethyl) phosphate (TBEP), triethyl phosphate, and *tris*(2-chloroethyl) phosphate [5]. In Beijing’s surface water, an average concentration of 14 kinds of OPFRs was 954 ng/L (3.24–10,945 ng/L). Among them, *tris*(1-chloro-2-propyl) phosphate (TCPP) and *tris*(chloroethyl) phosphate were identified as major chemicals [6].

OPFRs are assumed to be emitted from pollutants of household items and enter the influent water of the WWTP. Since it is detected mostly in WWTPs of residential areas and factory sewage, OPFRs in effluent remain and may be discharged into the environment, so research on OPFRs in WWTPs must be continued [7]. Although OPFRs have been identified in surface water and wastewater, studies on the removal process in the WWTPs are still insufficient [8]. Therefore, this study focused on OPFRs in WWTPs that perform biological treatment and the resulting transformation.

The microbial community of WWTPs contains various groups of organisms, most of which provide functional benefits such as water quality improvement [9]. Most WWTPs are currently designed to remove organic matter and nutrients in wastewater by passing through anaerobic, anoxic, and aerobic bioreactors [10]. In a previous study, pharmaceutical ibuprofen showed a high removal rate (90%) in WWTPs, resulting from absorption and microbial biodegradation [11]. In effluent, ibuprofen itself was rarely detected or appeared at low concentrations, but the biotransformation products (BTPs) hydroxyl ibuprofen and ibuprofen carboxylic acid were detected at much higher concentrations [12]. Fungicide triclosan was not only combined with methyl and hydroxyl molecules by microorganisms but also transformed into chlorophenol, which is a toxic substance. Intermediated products and final products generated by the chemical transformation in WWTPs might be more toxic than parent compounds [13]. Therefore, it is necessary to understand the biotransformation process of organic pollutants to optimize the removal efficiency of pollutants and the operation of the WWTP.

In this study, representative alkyl OPFR, chlorinated OPFR, and aryl OPFR were analyzed in the influent, biological reactor, and effluent of a WWTP. For each OPFR, the BTPs produced in the biological reactor were identified, and toxicity was predicted using a software tool. Microorganisms in WWTPs have the potential to transform the parent compound into another hazardous pollutant. Since BTPs discharged from the WWTP are introduced into the river and can affect the aquatic ecosystem, this study was conducted to predict a potential impact of BTPs on water environment.

## 2. Results

### 2.1. Quality Assurance and Quality Control (QAQC)

To increase the confidence of the analysis, the QAQC of OPFRs were confirmed. By exposing TBEP, TCPP, and TPHP to water, the average recovery was calculated by triplicate experiment at each concentration: TBEP 97% (93–105%), TCPP 103% (91–109%), TPHP 103% (84–111%). The method detection limit (MDL) was measured at least seven times at the same concentration. The MDL was found to be 0.08 µg/L in TBEP, 0.06 µg/L in TCPP, and 0.01 µg/L in TPHP.

### 2.2. Concentration of Organophosphate Flame Retardants in Wastewater Treatment Plant

OPFRs were analyzed at the Gwangju WWTP; the target sampling points, the concentration, and the elimination rate according to each location are shown in Table 1. In the influent, TBEP, TCPP, and TPHP were detected at concentrations of 2.55 μg/L, 0.13 μg/L, and 0.021 μg/L. In the effluent, OPFRs were found to be 1.09 μg/L, 0.27 μg/L, and 0.017 μg/L, respectively. The flame retardant which had the highest detected concentration in influent, biological reactor, and effluent was TBEP. Over 50% of TBEP and less 20% of TPHP in the influent were removed from WWTP by microorganisms. Rather, the concentration of TCPP increases as it undergoes biological treatment.

### 2.3. Identification of Biotransformation Products from Organophosphate Flame Retardants

In the study, only BTPs with clear chromatography peaks, isotopic patterns, and MS/MS fragments were selected. All BTPs were substances with an intensity of 10^4^ or higher. The Gwangju WWTP treats chemicals in wastewater by a biological method and discharges them into the river. During this process, OPFRs are highly likely to be transformed into BTPs. Therefore, this study was conducted focusing on the biotransformation of OPFRs by microorganisms. To identify BTPs caused by biological transformation, newly detected chemicals in biological reactors compared to influent were selected. Table 2 indicated the results of identifying BTPs of representative alkyl, chlorinated, and aryl OPFRs (TBEP, TCPP, TPHP) in positive/negative modes. Five types of BTPs were identified in TBEP, four in TCPP, and six in TPHP. Biotransformation of OPFRs was determined by referring to the literature and the EAWAG-BBD. The biotransformation reactions were mainly hydrolysis and oxidation, and in the case of TCPP, oxidative dechlorination occurred in which Cl-functional groups were removed.

Information on TBEP, TCPP, TPHP, and its BTPs identified in WWTP was shown in Table 2. The MS/MS fragment pattern of the major substances among each BTP are shown in Figure 1. All three BTPs were indicated in the form of oxidation from the OPFRs. Oxidation caused by organic pollutants is a potential toxicity mechanism in organisms. The oxidative BTPs formed by microorganisms could be discharged into the river by the effluent, so the study was required.

#### 2.3.1. *Tris*(2-butoxyethyl) Phosphate

Most of the five BTPs of TBEP were detected in the positive mode, and TB_88 was the only one identified in the negative mode. Figure 1a was the MS/MS fragment pattern of TB_414, the main product of TBEP. Information on the remaining BTPs is included in Figure 2. TB_414 was BTP generated from TBEP by oxidation of microorganisms. Peaks of the product were observed at 9.34 and 9.68 min (Table 2). Retention times pairing was due to the formation of an enantiomer in which an OH-functional group was bound to the parent compound. Precursor ion (C_18_H_39_O_8_P) was not identified in the MS/MS, but several fragments prove that an oxidative product was detected. Fragment C_4_H_8_O_4_P, C_7_H_18_O_4_P, C_9_H_22_O_4_P, and C_11_H_19_O_4_P are structures in which an alkyl or ether group was bound to a phosphorus oxide. In addition, a molecule of H_4_PO_4_ with H^+^ bound to phosphoric acid was identified at *m/z* 98.98410. The OH-functional group of TB_414 was transformed into a carbonyl group through oxidation, resulting in TB_412 (Figure 2d). Therefore, MS/MS fragment patterns of TB_412 and 414 were almost similar. In the MS/MS fragment of another product TB_88 generated by the oxidation, only the precursor ion (C4H7O2) was identified. It was judged that the mass information was insufficient due to the limitation of the small molecular weight. The oxidation that contributed to the formation of TB_88 was a different kind of reaction from TB_412 and TB_414; it was generated by aldehyde dehydrogenase from TBEP [14].

TB_298 was a hydrolysis BTP from TBEP by microbial esterase. TB_198 was generated after hydrolysis from TB_298 once more, and TB_298 and TB_198 had cognate MS/MS fragment patterns (Figure 2b,c). According to a previous study, exposure of TBEP to human liver microsomes generated hydroxylated TBEP and hydrolysis TBEP [15]. The result was shown to be the same as the BTPs of TBEP by microorganisms in this study.

#### 2.3.2. *Tris*(1-chloro-2-propyl) Phosphate

Chemicals that did not exist in influent and were detected in biological reactors and effluent were classified to identify BTPs generated by microorganisms. As a result, four kinds of BTPs of TCPP were identified in WWTP by microorganisms. Three were observed in the negative mode and one in the positive mode. The MS/MS fragment pattern of TC_290, the main BTP of TCPP, is described in Figure 1b. This was a form in which two chlorides in TCPP were converted into hydroxyl groups by oxidative dechlorination. BTPs were produced by dehalogenase in microorganisms present in the biological reactor [16]. *M/z* 184.01529 (C_4_H_9_O_6_P) was a fragment in which propyl chloride and methane were removed from TC_290. In addition, the presence of Cl was identified through fragments of CH2Cl and CH3OClP, and phosphate was identified by *m/z* 78.95898. Except for TC_249, TC_90 and TC_304 were BTPs were derived from TC_290 (Figure 3). TC_90 was formed by hydrolysis, and TC_304 by oxidation (Table 2). Only precursor ion (C_3_H_5_O_3_) was observed in the MS/MS fragment of TC_90 (Figure 3a). This was considered a lack of information due to the small molecular weight, similar to TB_88. In TC_304, the OH-functional group of TC_290 was oxidized by dehydrogenase and changed into a carboxyl group. The structure of TC_304 was predicted based on *m/z* 78.95881 (PO_3_), 119.03474 (C_4_H_7_O_3_), and 146.96076 (CH_5_O_4_ClP) (Figure 3c). Wan, et al. [17] confirmed the biotransformation reaction by exposing TCPP to wheat. The main reactions were oxidative dehalogenation and hydroxylation, which were also investigated in this study. Therefore, the significant biotransformations of TCPP were oxidative dechlorination and oxidation in organisms.

Halogenated compounds are persistently hazardous materials in the environment [18]. Halogenated compounds are generally non-degradable, but a larger degree of degradation occurs in anaerobic microorganisms than in aerobic microorganisms [19]. The cytochrome P450 enzyme in organisms has the potential to dehalogenate the halogenated hydrocarbons [20]. For that reason, in this study it is estimated that the oxidative dechlorination of TCPP was caused by cytochrome P450 in anaerobic microorganisms with the anaerobic process of the biological reactor.

#### 2.3.3. Triphenyl Phosphate

Six BTPs of TPHP generated by microorganisms were identified in the WWTP. Two types were detected in the positive mode, and four were detected in the negative mode. Among them, the MS/MS fragment pattern of TP_342, a major BTP, is presented in Figure 1c. TP_342 was a BTP formed by oxidation in microorganisms from TPHP. *M/z* 95.04924 (C_6_H_7_O) and 109.02821 (C_6_H_5_O_2_) were fragments that prove the phenyl groups of the chemical structure. *M/z* 124.99921 (C_2_H_6_O_4_P) and 153.06872 (C_5_H_14_O_3_P) were derived from the phosphate of TP_342. BTPs other than TP_342 were formed by hydrolysis in microorganisms. TP_250 was BTP generated only through hydrolysis reaction from TPHP (Table 2). This product has the same structure as diphenyl phosphate, and the results were confirmed by comparison with the MS/MS fragment pattern of the reference standard (Figure 4e). TP_250 and TP_342 were detected as the same BTPs in invertebrates exposed to TPHP [21]. It could be inferred that hydrolysis and oxidation might occur with the biotransformation reactions of TPHP regardless of species.

TP_110 had the same structure as hydroquinone, and the precursor ion C_6_H_5_O_2_ (*m/z* 109.02943) was confirmed in MS/MS (Figure 4a). TP_110 seemed to be produced by the degradation of TPHP by an enzyme (aryldialkylphosphatase) in microorganisms [22]. TP_138, an intermediate product between TP_188 and TP_140, was formed by hydrolysis and carboxylation from TPHP (Figure 4b).

### 2.4. Ecotoxicological Information of Biotransformation Products in Microorganisms

In this study, toxicity was predicted based on 15 kinds of BTPs formed by microorganisms in WWTP. The LC50 (green algae, daphnid, fish) of BTPs was determined by applying to the ECOSAR program from USEPA based on the chemical structure of detected BTPs. Figure 5 shows the biotransformation pathways and toxicity of OPFRs. The figure indicated on the structure of BTPs represents the value obtained by dividing 1,000,000 by the sum of the predicted LC50 for each species (10^6^/∑LC50). The LC50 for BTPs is described in Appendix A. In the experiment, the LC50 (EC50) of TBEP was found to be 61 mg/L (72 h) in green algae, 53 mg/L (48 h) in daphnid, and 24 mg/L (96 h) in freshwater fish [23]. The toxicity of TBEP was identified as 26–42 mg/L by ECOSAR, so the prediction data of the model and the experimental values were not significantly different (Appendix A). The experimental value of TCPP was 10–100 mg/L, and TPHP was 0.2–0.4 mg/L [24,25]. The other OPFRs had similar experimental and predicted values. Among the OPFRs in this study, TPHP seems to have the highest toxicity.

The toxicity of BTPs predicted by the ECOSAR model was mostly reduced compared to the parent compound. In TBEP, the toxicity intensity was estimated to be 10^4^, while other BTPs were observed to be less than about 10^3^ (Figure 5a). In TCPP, the toxicity of BTPs decreased sharply compared to other OPFRs (Figure 5b). Except for TC_249, the toxicity of BTPs of TCPP was expected to decrease by more than 100 times compared to the parent compound. On the other hand of TPHP, transformations were carried out in the form of decreasing or increasing product toxicity (Figure 5c). The BTP with the hydroxyl group (TP_342) showed the same intensity of toxicity as the parent compound. TP_110 was hydroquinone and was expected to exhibit higher toxicity than TPHP.

## 3. Discussion

As a result of this study, the OPFR with the highest concentration in the influent was TBEP. The production of TBEP is 5000–6000 tons per year worldwide, and a large amount is used as a substitute for brominated flame retardants [26]. TBEP is an additive that does not chemically bond to the product, so it could be easily released into the water [4]. Due to these characteristics, it is believed that the concentration of TBEP entering the WWTP was high. It was detected in the order of TBEP, TCPP, and TPHP in the influent, and the concentration of TCPP was rather increased in the effluent. Similar results have been published in the previous study. OPFRs were detected in the German Ruhr/Rhine regional sewage treatment plant; TBEP (3.7 μg/L in influent and 0.4 μg/L in effluent), TCPP (2 μg/L and 3 μg/L), and TPHP (0.13 μg/L and 0.07 μg/L) [7]. The reason that the concentration of TCPP increased in WWTP effluent was that the biocube used in the aerobic tank. During the biological treatment stage of the Gwangju WWTP, the aerobic tank uses a biocube (or biomedia) to increase the number of microorganisms to treat wastewater. The biocube applied in the A_2_O (anaerobic/anoxic/aerobic) process is a carrier made of polyurethane to contain microorganisms, and TCPP is a major flame retardant added to polyurethane production [27]. Because there is a large amount of biocube to supply microorganisms in the aerobic tank, it is expected that TCPP was eluted into the wastewater.

Oxidative BTPs of OPFRs were produced by microorganisms in WWTP. The hydroxyl groups of BTPs have the potential to cause toxicity. According to a previous study, *Escherichia coli* exposed to polybrominated diphenyl ether (PBDE) generated hydroxylated brominated diphenyl ether (HO-BDE) through biotransformation in the body. The hydroxylated BTP inhibited the growth of *E. coli* and interfered with the metabolic balance [28]. The reason that oxidative BTPs increase toxicity, as well as genomic and mutational effect, is that the hydrophilic OH-functional groups increased water solubility and biological reactivity [29].

Organic pollutants are transformed into BTPs through biological reactions, and the products might have different toxicity compared to the parent compound [30]. The type and intensity of toxicity reactions exhibited by chemicals are regulated by the characteristics and extent of biotransformation [31]. In general, biological transformation reaction converts a chemical into a less polarized and more easily excreted form than the parent compound. In addition, the biotransformed products via phase I and phase II are generally more hydrophilic and less toxic than the hydrophobic and toxic parent compounds. However, some biotransformation could generate products that are much more toxic than the parent compound [32,33]. In this study, hydroquinone in particular was predicted to have higher toxicity than the parent compound TPHP. Hydroquinone is a stable compound with various resonance structures, so it could be highly toxic as well as soluble. Hydroquinone must be carefully monitored in the environment, as prolonged exposure to aquatic organisms could cause serious toxicity [34]. When BTPs generated from organic pollutants are stable, or the biotransformation rate is slower than the parent compound, BTPs can be continuously produced by the transformation/decomposition process in a WWTP [8]. Since the BTPs produced in this way have the potential to flow into rivers, continuous research on biotransformation is required.

## 4. Materials and Methods

### 4.1. Chemicals and Reagents

Each type of alkyl OPFR, chlorinated OPFR, and aryl OPFR was prepared for the sample analysis of the WWTP. *Tris*(2-butoxyethyl) phosphate (TBEP), *tris*(1-chloro-2-propyl) phosphate (TCPP), and triphenyl phosphate (TPHP) were purchased from Sigma-Aldrich (St. Louis, MO, USA). Internal standard (TPHP-*d*_15_, 98 atom% D) and formic acid added to the mobile phase were also prepared by Sigma-Aldrich. Methanol and water (HPLC grade) were used as the mobile phase for analysis (Thermo Fisher Scientific: St. Waltham, MA, USA).

### 4.2. Information on the Wastewater Treatment Plant

Samples were collected in April 2020 at the first WWTP in Gwangju, South Korea (35°09′22.7” N 126°49′51.8” E). The source of inflow to the WWTP is municipal and industrial wastewater. The facility treats 600,000 tons of wastewater per day and covers an area of about 450,000 m^2^, and the discharged water flows into the Youngsan River. This WWTP takes steps to remove contaminants through biological methods, so it is suitable for identifying the transformation in OPFRs (Appendix A). Although the activity of microorganism in bioreactor might be variable due to different organic load with time, samples were collected at three sites in duplicate: influent, immediately after the biological reactor, and effluent.

### 4.3. Sample Preparation

Water from the WWTP was filtered using a 125-mm GF/C microfiber filter within 12 h of collecting to minimize changes in the sample. Then, 100 μL internal standard was injected into the sample, and the final volume was adjusted to 1 L. The sample was extracted using solid-phase extraction (SPE: AutoTrace 280, Thermo Scientific; Waltham, MA, USA) to identify OPFRs and their biotransformation products. Next, 6-mL, 500-mg Oasis HLB cartridges (Waters: Milford, MA, USA) were conditioned with 5 mL water and methanol, respectively. The samples were loaded at 8 mL/min, and then the cartridge was dried with nitrogen gas for 40 min. It was eluted to a final volume of 10 mL using methanol. A nitrogen concentrator (Hurricane-lite, Chongmin tech: Gangseo-gu, Seoul, Korea) was used to concentrate the sample volume for analysis to 1 mL. Samples were stored in a deep freezer until analysis.

### 4.4. Chemical Analysis

An ultra-high-performance liquid chromatography-electrospray ionization-quadrupole orbitrap mass spectrometer (UHPLC-HRMS/MS, Thermo Fisher Scientific Inc: Waltham, MA, USA) was used to analyze OPFRs and biotransformation products in the WWTP. The mobile phase was water (A) and methanol (B), each containing 0.1% formic acid. The sample was passed through an Xbridge C18 column (2.1 × 50 mm, 3.5 µm: Waters, Milford, MA, USA) to separate the chemicals. The UHPLC gradient is described in Appendix A. Orbitrap was analyzed data-dependent MS/MS fragment for five highest intensive ions in full scan. The range of full scan was from 50 to 750 *m/z*, mass resolution of 140,000, and 10 ppm of mass accuracy. Analysis conditions were as follows: 10 μL injection volume, 45 L/min sheath gas, 10 L/min auxiliary gas, and 320 °C capillary temperature. Samples were analyzed in both positive and negative modes: source voltages were 3.8 kV in the positive mode and -3 kV in the negative mode.

Quantitative analysis of OPFRs was performed with standard compounds, and qualitative analysis of BTPs without standard was identified through a prediction process. The transformation pathway of OPFRs was estimated by the EAWAG-BBD (Biocatalysis/Biodegradation Database) pathway prediction system (http://eawag-bbd.ethz.ch/predict/, accessed on 12 October 2020). CompoundDiscoverer 2.0 (Thermo Scientific: Waltham, MA, USA) was also used to predict degradation products from OPFRs. When information existed as a reference standard among BTPs, it was confirmed by comparing the mass spectral database (*m/z* Cloud: Waltham, MA, USA).

### 4.5. Toxicity Prediction

The ecological structure activity relationships (ECOSAR) predictive model was used to predict the toxicity of BTPs by microorganisms as well as OPFRs. ECOSAR estimates the toxicity of aquatic organisms based on the similarity of the chemical structure from the USEPA database. For this, ECOSAR V1.11 distributed by USEPA was applied (https://www.epa.gov/tsca-screening-tools/ecological-structure-activity-relationships-ecosar-predictive-model, accessed on 15 March 2021). From the data, we selected the acute toxicity values (LC50) of green algae (96 h), daphnid (48 h), and freshwater fish (96 h).

## 5. Conclusions

In this study, the concentration of OPFRs (TBEP, TCPP, and TPHP) and the removal rate of organic pollutants were identified according to the wastewater treatment process. As a result, TBEP was found to have the highest concentration, and TCPP was not eliminated by the biological reaction of the WWTP. In the case of TCPP, the concentration was higher in the effluent than in the influent due to the biocube component of the biological reactors. OPFRs introduced into the WWTP were transformed into various biotransformation molecules by anaerobic and aerobic microorganisms. Five biotransformation types of TBEP, four of TCPP, and six of TPHP were generated by microorganisms. Common microbial biotransformation of TBEP, TCPP, and TPHP was observed as hydrolysis and oxidation. The resulting BTPs were generally predicted to have lower toxicity than the parent compound, but products such as hydroquinone were predicted to be highly toxic. The BTPs are mixed with the effluent and flow into the river, which has the potential ultimately to affect the aquatic ecosystem. The BTPs of OPFRs identified in this study could be cited as data for screening chemicals in water. In addition, when BTPs against OPFRs are detected in rivers, it is possible to predict whether they are derived from biological reactions in WWTPs. Considering that the removal rate of OPFRs in the wastewater treatment plant was low and other biotransformation molecules were generated, the wastewater treatment technology needs to be changed. It was confirmed that the chlorinated biotransformation molecule was formed through the biocube. In addition, toxic substances could affect microorganisms in biological reactors. In order to efficiently remove organic pollutants, it is necessary to apply an alternative and additional technique other than biological treatment.

## Figures and Tables

**Figure 1 ijms-22-05376-f001:**
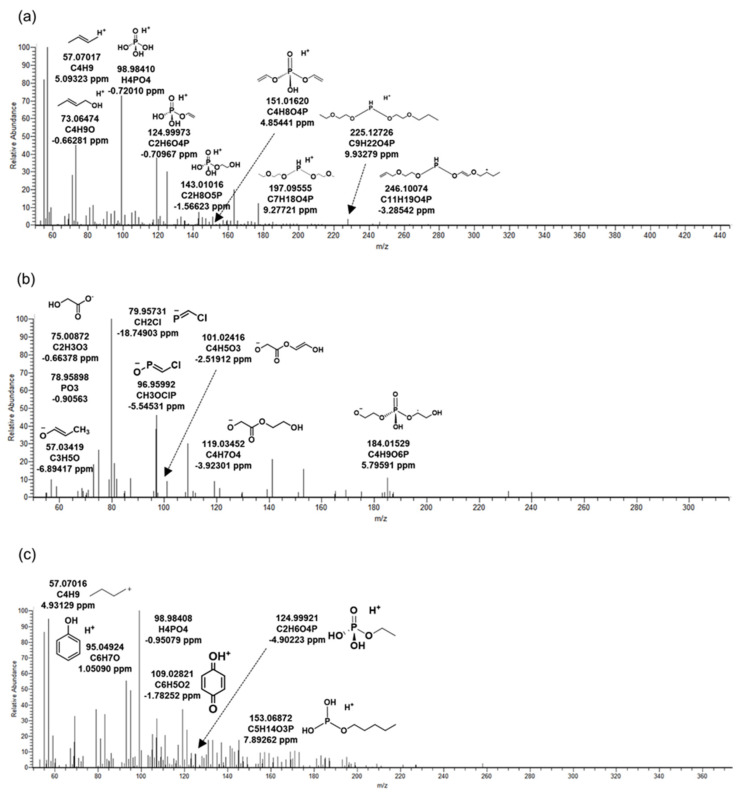
MS/MS fragments at oxidative biotransformation products of organophosphate flame retardants generated in biological reactor from wastewater treatment plant. (**a**) TB_414: biotransformation product (BTP) of *tris*(2-butoxyethyl) phosphate, (**b**) TC_290: BTP of *tris*(1-chloro-2-propyl) phosphate, (**c**) TP_342: BTP of triphenyl phosphate.

**Figure 2 ijms-22-05376-f002:**
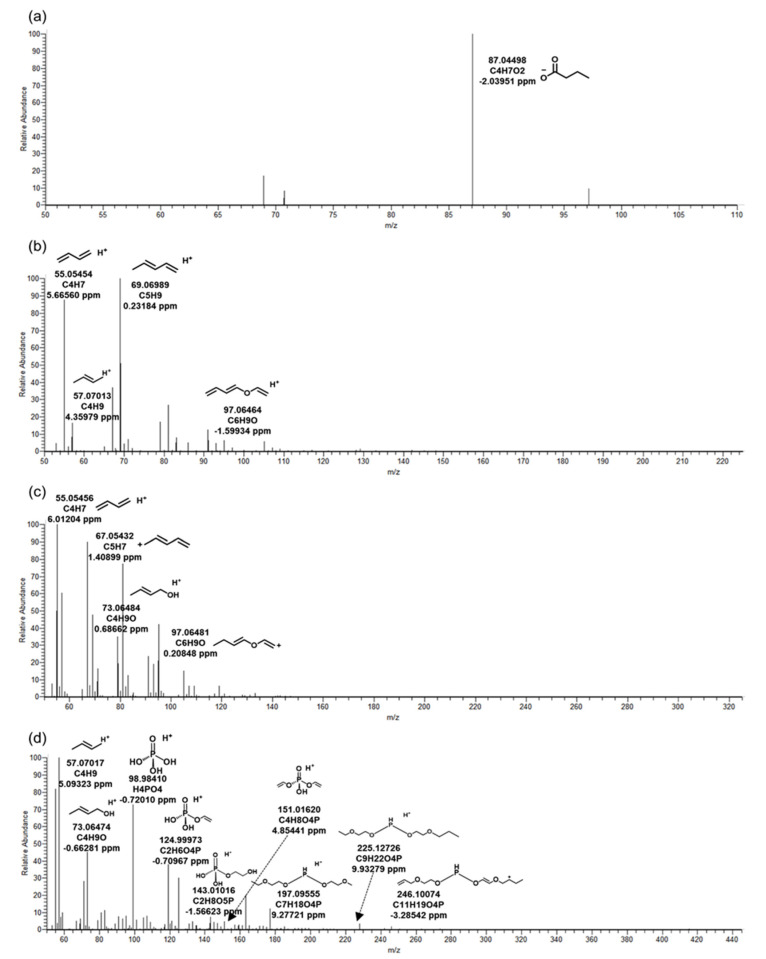
MS/MS fragments of *tris*(2-butoxyethyl) phosphate in biological reactor of wastewater treatment plant. The biotransformation products: (**a**) TB_88, (**b**) TB_198, (**c**) TB_298, (**d**) TB_412.

**Figure 3 ijms-22-05376-f003:**
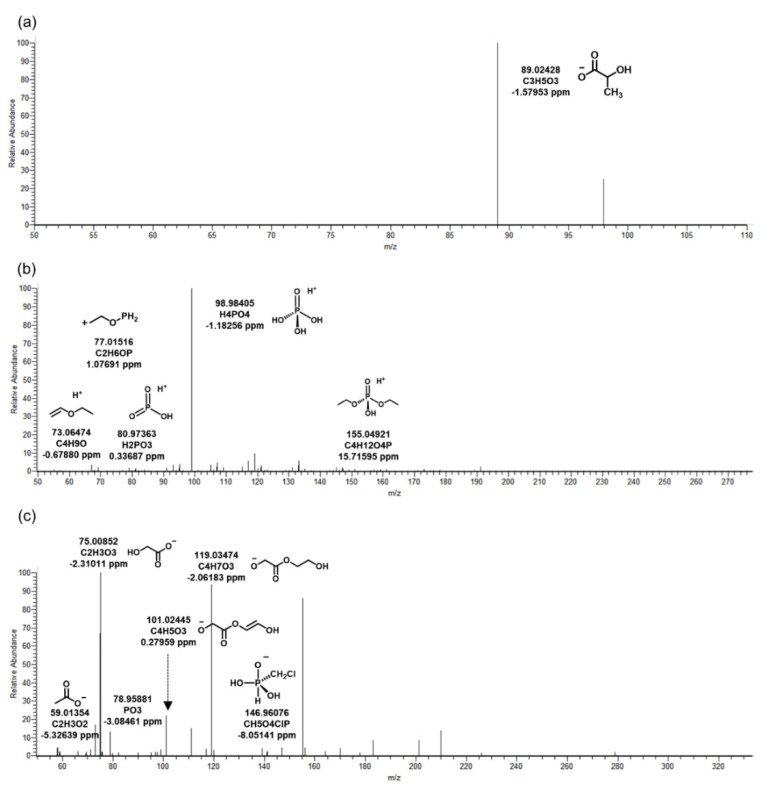
MS/MS fragments of *tris*(1-chloro-2-propyl) phosphate in biological reactor of wastewater treatment plant. The biotransformation products: (**a**) TC_90, (**b**) TC_249, (**c**) TC_304.

**Figure 4 ijms-22-05376-f004:**
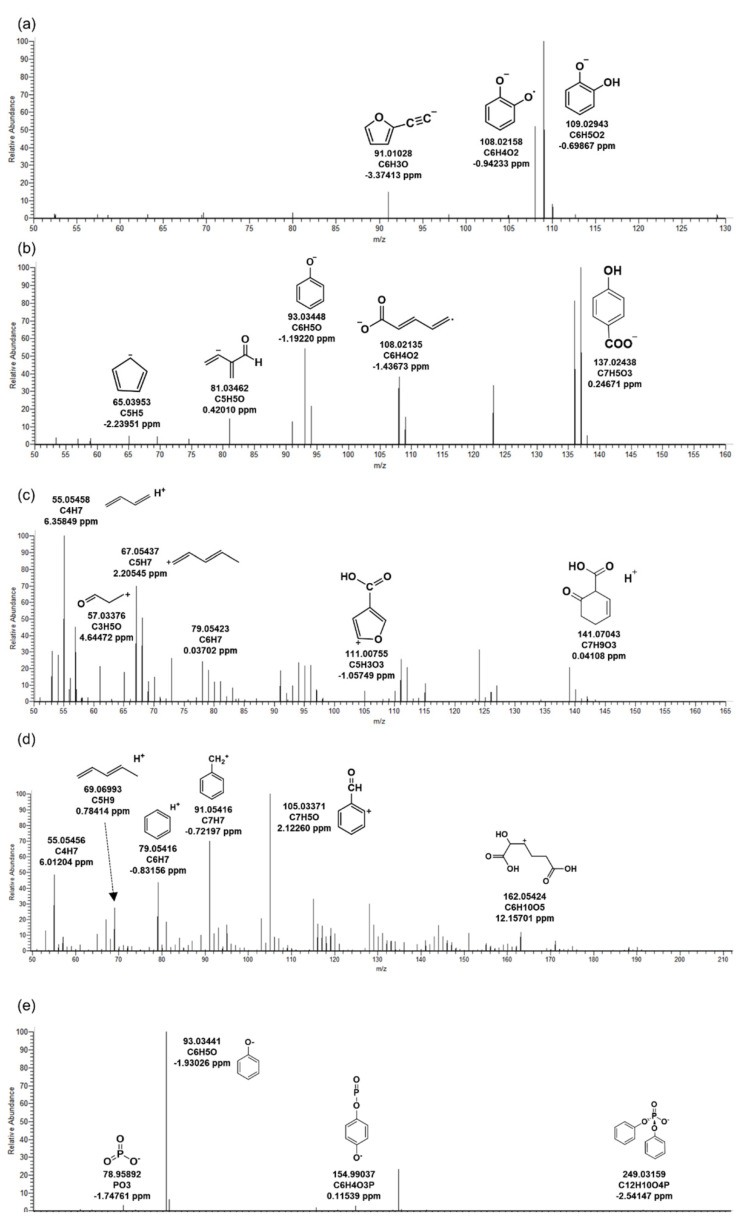
MS/MS fragments of triphenyl phosphate in biological reactor of wastewater treatment plant. The biotransformation products: (**a**) TP_110, (**b**) TP_138, (**c**) TP_140, (**d**) TP_188, (**e**) TP_250.

**Figure 5 ijms-22-05376-f005:**
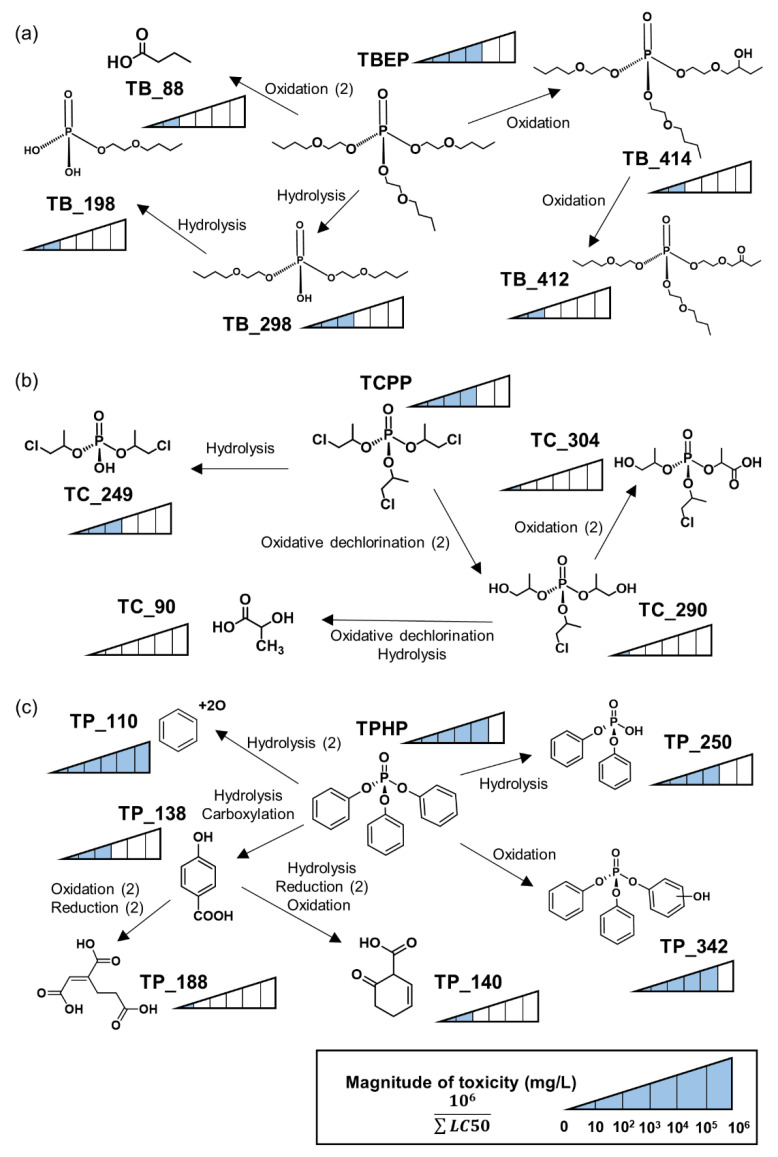
Biotransformation pathway and predicted toxicity of (**a**) *tris*(2-butoxyethyl) phosphate, (**b**) *tris*(1-chloro-2-propyl) phosphate, (**c**) triphenyl phosphate. The arrow represents the pathway to the formation of biotransformation products. The number of blocks indicates the estimated magnitude of the toxicity of substances.

**Table 1 ijms-22-05376-t001:** The concentration of OPFRs in the influent, biological reactor, and effluent of the wastewater treatment plant. The following OPFRs were analyzed: TBEP (*tris*(2-butoxyethyl) phosphate), TCPP (*tris*(1-chloro-2-propyl) phosphate), and TPHP (triphenyl phosphate). When the removal rate of chemical was less than 0, it was expressed as none.

OPFRs	Influent [μg/L]	Biological Reactor [μg/L]	Effluent [μg/L]	Removal Rate (%)
TBEP	2.55	1.24	1.09	57.3
TCPP	0.13	0.24	0.27	None
TPHP	0.021	0.019	0.017	19.0

**Table 2 ijms-22-05376-t002:** Information on organophosphate flame retardants (OPFRs) and their biotransformation products by microorganisms in the wastewater treatment plant (WWTP).

Compound ^a^	Formula	Mass of [M+H]^+^ or [M-H]^−^	Retention Time (min)	Formula Change	Biotransformation
TBEP	C_18_H_39_O_7_P	[M + H]^+^: 399.25062	12.76		Parent compound
TB_88	C_4_H_8_O_2_	[M − H]^−^: 87.04515	9.26	− C_12_H_31_O_5_P	Oxidation (2)
TB_198	C_6_H_15_O_5_P	[M + H]^+^: 199.07299	5.66	− C_12_H_24_O_2_	Hydrolysis (2)
TB_298	C_12_H_27_O_6_P	[M + H]^+^: 299.16180	11.15	− C_6_H_12_O	Hydrolysis
TB_412	C_18_H_37_O_8_P	[M + H]^+^: 413.22988	9.38	− H_2_ + O	Oxidation (2)
TB_414	C_18_H_39_O_8_P	[M + H]^+^: 415.24553	9.34, 9.68	+ O	Oxidation
TCPP	C_9_H_18_Cl_3_O_4_P	[M + H]^+^: 327.00811	8.62		Parent compound
TC_90	C_3_H_6_O_3_	[M − H]^−^: 89.02441	16.45	− C_6_H_12_Cl_3_OP	Oxidative dechlorination (3),Hydrolysis
TC_249	C_6_H_13_Cl_2_O_4_P	[M + H]^+^: 251.00013	8.64	− C_3_H_5_Cl	Hydrolysis
TC_290	C_9_H_20_ClO_6_P	[M − H]^−^: 289.06132	4.95	− Cl_2_ + H_2_O_2_	Oxidative dechlorination (2)
TC_304	C_9_H_18_ClO_7_P	[M − H]^−^: 303.04059	9.08	− Cl_2_ + O_3_	Oxidative dechlorination (2),Oxidation (2)
TPHP	C_18_H_15_O_4_P	[M + H]^+^: 327.07807	10.89		Parent compound
TP_110	C_6_H_6_O_2_	[M − H]^−^: 109.02950	3.61	− C_12_H_9_O_2_P	Hydrolysis (2)
TP_138	C_7_H_6_O_3_	[M − H]^−^: 137.02440	2.47	− C_11_H_9_OP	Hydrolysis, Carboxylation
TP_140	C_7_H_8_O_3_	[M + H]^+^: 141.05460	2.48	− C_11_H_7_OP	Hydrolysis (2), Reduction (2),Oxidation, Carboxylation
TP_188	C_7_H_8_O_6_	[M + H]^+^: 189.03940	4.98	− C_11_H_7_P + O_2_	Hydrolysis, Oxidation (2), Reduction (2), Carboxylation
TP_250	C_12_H_11_O_4_P	[M − H]^−^: 249.03222	6.27	− C_6_H_4_	Hydrolysis
TP_342	C_18_H_15_O_5_P	[M + H]^+^: 343.07299	9.01	+ O	Oxidation

^a^ Abbreviation: TBEP (*tris*(2-butoxyethyl) phosphate), TCPP (*tris*(1-chloro-2-propyl) phosphate), TPHP (triphenyl phosphate). TB, TC, and TP stand for the biotransformation products of TBEP, TCPP, and TPHP, respectively. The following number represents the neutral mass of the product.

## Data Availability

Not applicable.

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
