# Peer review of "Identification and Toxicity Prediction of Biotransformation Molecules of Organophosphate Flame Retardants by Microbial Reactions in a Wastewater Treatment Plant"

_ijms, 2021, doi:10.3390/ijms22105376_

Round 1
Reviewer 1 Report
In this study, OPFRs introduced into a wastewater treatment plant (WWTP) were identified, and the toxicity of biotransformation molecules generated by the biological reaction was predicted.
It is an interesting study, however I have some comments, before continuing with its processing:
Page 10
In figure 5, indicate if it is an oxidation, hydrolysis, dehalogenation or reaction that corresponds to each arrow. With the help of chromatograms, the percentages of each compound in which the initial organophosphates are degraded can be recorded.
Although gas / mass chromatography is a very efficient technique, an FTIR spectrum would have been very useful, since it would have given information on the appearance and disappearance of functional groups, which would further reinforce the part where all the by-products of the degradation of the initial organophosphates.
Rule 237 to 218
Reinforce the toxicity part by mentioning that this can not only be defined by the structure of the molecules, but also by the functional groups they present, as well as their stability, this stability can be by alternating double bonds or by electron density that confer functional groups on molecules. These characteristics promote greater or lesser solubility and toxicity of these compounds.
Line 241 to 257
Because they did not take COD and BOD into account, since, due to the by-products generated, it is very likely that these parameters have increased and most likely their influent is more toxic than the effluent.
Reverse the order, first methodology and then results and discussion.
Line 340 to 355
In the conclusions part, emphasis should be placed on the low efficiency of the treatment and make recommendations to improve the process.
It would be important from the beginning to indicate that it is a toxic effluent and that therefore a biological treatment is not very efficient.
Line 263 to 265
Mention that the OH group are hydrophilic and therefore increases the solubility of these compounds in water. Talk about the hydrophobic and hydrophilic properties of organic compounds.
Methodology
It is not clear how many samples were obtained, if it was only punctual and in a single schedule, the data would not be giving clear information for the whole day, remember that there is variation in time, as well as in the organic load. So it is advisable to monitor at least a full day in time intervals (morning, noon, afternoon and night).
Reviewer 2 Report
See attached file

Round 2
Reviewer 1 Report
The comments were taken care of and I consider that the manuscript can be accepted in its current version.
